# A shape model of internally mixed soot particles derived from artificial surface tension

Hiroshi Ishimoto[1], Rei Kudo[1], Kouji Adachi[1]

[1]Meteorological Research Institute, Tsukuba, 305-0052, Japan

*Correspondence to*: Hiroshi Ishimoto (hiroishi@mri-jma.go.jp)

**Abstract.**

To retrieve the physical properties of aerosols from multi-channel ground-based/satellite measurements, we developed a shape model of coated soot particles and created a dataset of their optical properties. Bare soot particles were assumed to have an aggregate shape, and two types of aggregates with different size–shape dependences were modeled using a polyhedral Voronoi

structure. To simulate the detailed shape properties of mixtures of soot aggregates and adhered water-soluble substances, we propose a simple model of surface tension derived from the artificial surface potential. The light-scattering properties of the modeled particles with different volume fractions of water-soluble material were calculated using the finite-difference time-domain method and discrete-dipole approximation. The results of the single-scattering albedo and asymmetry factors were compared to those of conventional internally mixed spheres (i.e., effective medium spheres based on the Maxwell–Garnett

approximation and simple core–shell spheres). In addition, the lidar backscattering properties (i.e., lidar ratios and linear depolarization ratios) of the modeled soot particles were investigated. For internally mixed soot particles, the lidar backscattering properties were sensitive to the shape of the soot particles and the volume mixing ratio of the assumed water-soluble components. However, the average optical properties of biomass smoke, which have been reported from in situ field and laboratory measurements, were difficult to explain based on the individually modeled particle. Nonetheless, our shape

model and its calculated optical properties are expected to be useful as an alternative model for biomass smoke particles in advanced remote sensing via multi-channel radiometer and lidar measurements.

## 1 Introduction

During the atmosphere aging process of emitted combustion products, soot particles tend to become hydrophilic and form mixtures with weakly light-absorbing materials (Mikhailov et al., 2006)(Adachi et al., 2007)(Moteki and Kondo, 2007)

(Shiraiwa et al., 2007)(Adachi and Buseck, 2008)(Shiraiwa et al., 2010). Because of significant enhancements in light absorption and scattering, it has been suggested that soot particles in a mixing state are the second-most important contributor to global warming after carbon dioxide (Jacobson, 2001)(Ramanathan et al., 2008). For climate monitoring and numerical prediction via atmospheric data assimilation, precise estimation of the amount of mixed soot particles and the fraction of soot (black carbon) from satellite-/ground-based remote-sensing measurements is important (Kahnert et al., 2013). Thus,

understanding the optical properties of internally mixed soot particles is essential to improve the retrieval accuracy of atmospheric soot particles (Hara et al., 2018). The light-scattering properties of internally mixed particles depend strongly on the complex refractive index of each mixing component. Furthermore, the shape of the incorporated soot particles and overall particle shape in the mixing state significantly alter some of the scattering properties. In particular, particle shape is important for the interpretation of lidar backscattering measurements. Many shape models have been proposed for internally mixed soot

particles and their light-scattering properties using the discrete-dipole approximation (DDA) and T-Matrix methods (Adachi et al., 2010)(Scarnato et al., 2012)(Cheng et al., 2014)(Dong et al., 2015)(Liu et al., 2016)(Mishchenko et al., 2016) (Moteki, 2016) (Wu et al., 2016)(Wu et al., 2017) (Kahnert, 2017) (Zhang et al., 2017) (Luo et al., 2018). However, the relationship between mixing state/morphology and light-scattering properties is not well defined.

As an alternative model for soot particles in a mixing state, we developed a new shape model of internally mixed particles. We assumed that bare soot particles with fractal-like shapes were mixed with water-soluble (WS) components. Moreover, we considered hydrophilic soot particles with high wettability due to atmospheric aging. The shapes of the mixed particles were determined by applying artificial potential calculations of the surface tension of the WS components. The numerical results of the light-scattering properties of the modeled particles at visible and near-infrared wavelengths are discussed.

## 2 Shape model of internally mixed aerosols

### 2.1 Soot model

Bare soot particles are commonly described as fractal aggregates formed from primary particles (or monomers) with a degree of overlapping and necking between neighboring primary particles (Yon et al., 2015)(Okyay et al., 2016). Primary particles have a diameter of 20–50 nm (Bond and Bergstrom, 2006), and the fractal dimension of the aggregates depends on emissions conditions and atmospheric aging. For example, newly generated soot aggregates often form lace-like structures with relatively small fractal dimensions, whereas aged soot aggregates tend to be compact and to be characterized by large fractal dimensions (Mikhailov et al., 2001)(Mikhailov et al., 2006)(Zhang et al., 2016). In this study, we modeled soot aggregates using spatial Poisson–Voronoi tessellation. The basic methods to make the aggregate model are the same as those of dust particles and ice aggregates described in our previous works (Ishimoto et al., 2010)(Ishimoto et al., 2012a)(Ishimoto et al., 2012b) (Baran et al., 2018). A spatial Poisson–Voronoi tessellation was produced from randomly distributed nucleation points in the numerical field, and polyhedral cells overlapping with the assumed fractal frame were selected from the tessellation (Ishimoto et al., 2012a). We regarded the cells (i.e., Voronoi cells) as the primary soot aggregate particles to mimic overlapping and necking between neighboring primary particles. To ensure that the size of the Voronoi cells was relatively uniform, we applied a Matérn hard-core point field (Ohser and Mücklich, 2000) to the spatial distribution of the nucleation points (Ishimoto et al., 2012b). Aggregate particles of different sizes were produced in the same manner but by changing the relative size of the fractal frame within the same tessellation. Figure 1 shows two sets of aggregate particles (Types A and B) created from fractal frames of different shapes. To calculate the light-scattering properties, Type A aggregates of 10 sizes and Type B aggregates of 13 sizes (where Nos. 1, 2, and 3 were used for both sets) were prepared. The mean radius of the primary particle $a = 20$ nm was used as a typical value (Wu et al., 2015) (Wu et al., 2017)(Mishchenko et al., 2016)(Luo et al., 2018), and the total size of each aggregate was corrected by adjusting the average cell volume of the aggregate as $V_{c,\text{agg}} = 4/3\pi a^3$. Although the aggregates shown in Figure 1 were created using a box-counting approach to maintain a fractal relationship between size and shape, this fractal relationship differed from that of the commonly used fractal dimension $D_f$ used to describe individual soot particles. The fractal dimension $D_f$ is defined from the number of monomers $N$, fractal prefactor $k_0$, and gyration radius $R_g$ as follows (Adachi et al., 2007)

$$N = k_0 \left(\frac{R_g}{a}\right)^{D_f}. \qquad (1)$$

For numerical simulation of the light-scattering properties of aged soot particles, various $D_f$ values for a typical prefactor of $k_0 = 1.2$ have been proposed in the literature, such as $D_f = 2.0 - 2.5$ (Nyeki and Colbeck, 1995), $1.9 - 2.6$ (Adachi et al., 2007), 2.5 (He et al., 2015), 2.6 (Mishchenko et al., 2016), and $2.5 - 3.0$ (Zhang et al., 2017). The corresponding $D_f$ values for our modeled soot particles derived from the calculation of $R_g$ are plotted in Figure 2. Applying the fractal prefactor $k_0 = 1.2$ resulted in aggregate particles in the range of $D_f = 1.9 - 2.5$ (Type A) and $2.5 - 3.0$ (Type B) given a normalized gyration radius of $\ln(R_g/a) \leq 3.6$.

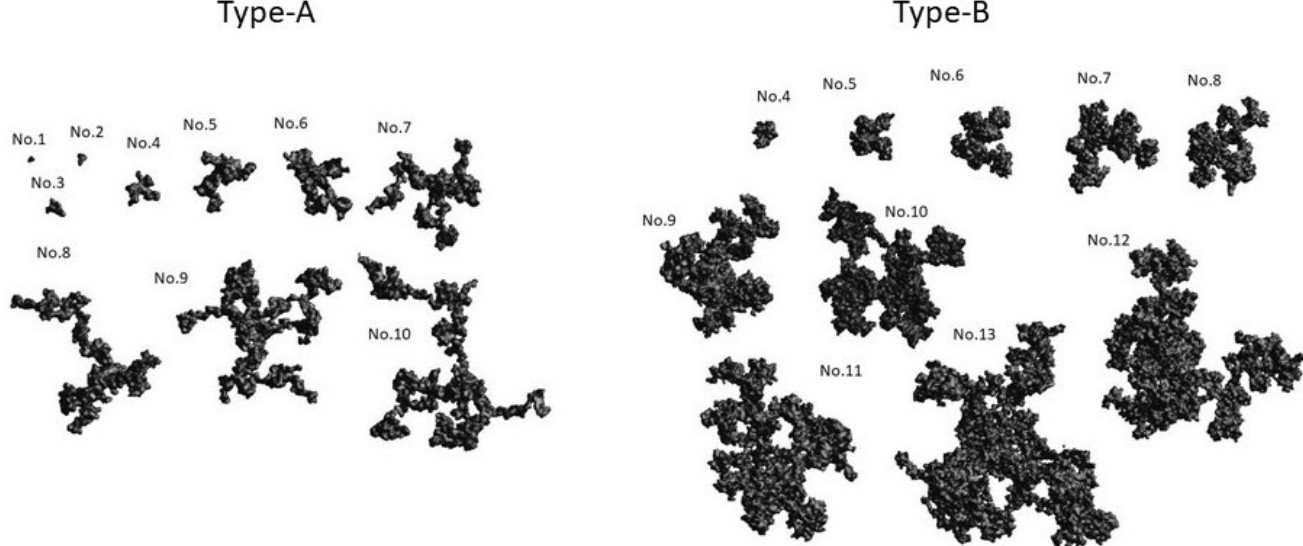

Type-A

Type-B

Figure 1. Model of bare soot particles created using three-dimensional Voronoi tessellation (Types A and B).

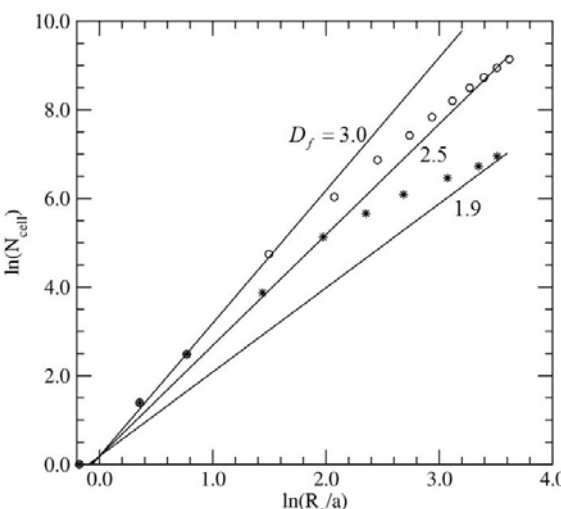

Figure 2. Relationship between the number of cells $N_{cell}$ (assuming $N_{cell} = N$, from Eq. (1) and normalized gyration radius $R_g/a$ for the modeled soot aggregates. Asterisks and open circles correspond to Type A and Type B aggregates, respectively. Solid lines represent the relationship described by Eq. (1) for fractal dimensions of $D_f = 1.9, 2.5, 3.0$ when $k_0 = 1.2$.

## 2.2 Artificial surface tension of mixed soot and water-soluble components

According to microscopic images of internally mixed soot particles, soot particles are often entirely encapsulated in a spherical shell and completely covered by WS components (Reid and Hobbs, 1998)(Reid et al., 2005a). Thus, WS components behave like a liquid on the particle surface, and the surface tension of WS components is important to describing the overall shapes of

mixed particles. Surface tension is the result of inter-molecular forces at the microscopic scale, described as the Lennard–Jones potential (Becker et al., 2014). Although some approaches to simulate surface tension in a discrete particle system, such as smoothed particle hydrodynamics (SPH), have been proposed (Tartakovsky and Meakin, 2005) (Leinonen and von Lerber, 2018), a general approach to reproduce the variety of fluid effects has not yet been developed (Akinci et al., 2013). In this study, we examined a virtual and simple potential field on the surface of the modeled particle to simulate the morphological effects of surface tension. We assumed that the dynamic behavior of liquid could be simulated by the movement of liquid elements from locations of high potential to locations of low potential. From a simple molecular-scale explanation of surface tension, the potential at a surface point becomes smaller as the number of surrounding molecules increases. The modeled particles were projected in a three-dimensional Cartesian grid space to define surface points for the potential calculations. The artificial surface potential was defined as an analog of the microscopic surface potential.

$$U_j \equiv -\sum_i f_i \,, \qquad \begin{cases} f_i = 1 & (r_{ij} \le d) \\ f_i = 0 & (r_{ij} > d) \end{cases} \qquad (2)$$

where $U_j$ is the artificial potential at surface point $j$, and $U_j$ is defined as the negative value of the total number of grid points of material with distance $r \le d$. We used this simple model for the surface potential calculations to focus on the shape of the mixture in discrete grid space. An equilibrium mixing state was simulated assuming that the applied WS components preferentially accumulate at grid points of lower potential. The shape of internally mixed particles under an arbitrary volume mixing ratio was determined based on iterative calculations of the surface potential and by adding WS elements. In the artificial potential calculations in Eq. (2), setting of the length $d$ is important. We applied two steps for the potential calculations and adhesion of WS components for each iterative calculation.

$$d_1 = 3l \,, \qquad N_{\mathrm{add}} = 0.016 N_s$$
$$d_2 = Max(d_1, d_{cor}) \,, \quad N_{\mathrm{add}} = 0.004 N_s, \qquad (3)$$

where $l$ is the grid length of the space, $d_1$ and $d_2$ are the lengths $d$ for first and second steps, and $N_{\mathrm{add}}$ is the number of grid points for WS adhesion, which are chosen from the total surface points $N_s$ (note that a surface point is defined as an empty grid and neighbor of an occupied grid). Small $d_1$ values resulted in a coated particle with a thin layer at a local scale. By contrast, WS components tended to accumulate in the same region on particles when a large $d_1$ was applied. We assumed the parameter $d_1$ to be the minimum scale to derive isotropic potential in the discrete Cartesian grid space and determined the value $d_1 = 3l$ from our results for shapes of coated particles in preliminary calculations. The second step in Eq. (3) is important, particularly when WS components cover the entire soot aggregate, and the value $d_2$ is determined to ensure that the overall shape of the mixed particle is spherical. The correction length $d_{cor}$ is the minimum length at which the curvature of the sphere with a radius $R$ can be discriminated from the calculated artificial potential in the grid field, and $d_{cor}$ is estimated from the following relationships.

$$\tfrac{4}{3}\pi R^3 = N_V l^3 \,, \quad 4\pi R^2 \gamma = N_s l^3, \quad R \sin\theta = d_{cor} \,, \quad R \cos\theta = R - l \qquad (4)$$

$$d_{cor} = l \sqrt{\frac{6 N_V \gamma}{N_s l} - 1} \qquad (5)$$

where $N_V$ is the total number of grid points occupied by the material, and $\gamma$ is the effective skin depth ($\gamma = 1.51l$). Figure 3 presents a schematic diagram describing $d_{cor}$ in a two-dimensional case.

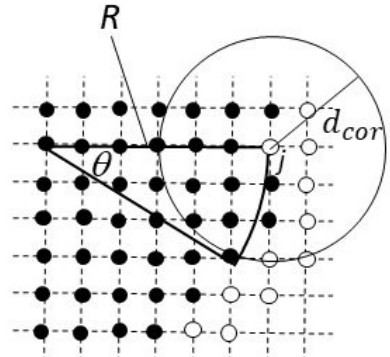

**Figure 3. Schematic diagram of the distance $d_{cor}$ based on the artificial potential calculation for a two-dimensional case. Open circles represent surface points for the potential field calculation, and solid circles indicate points occupied by material. To discriminate the curvature of the radius $R$ from the calculated potential at surface point $j$, the length $d$ in Eq. (2) should be larger than $d_{cor}$. A spherical mixed particle is automatically generated from the potential calculations and adding water-soluble material when the soot particle is completely encapsulated.**

The artificial potential was calculated for each iterative step, and $N_{add}$ of WS elements was added starting with the grid points of lowest potential. The internally mixed soot model (i.e., the particle shape modeled based on artificial surface tension, AST hereafter) was created for different values of the volume ratio $V_r = V_{ws}/V_{soot}$, where $V_{ws}$ and $V_{soot}$ are the volumes of the WS and soot materials, respectively. Figure 4 shows the results of several mixed soot models for $V_r \sim 0, 2, 5, 10, 20$. The total number of iterations was approximately 1000 (2000) to create $V_r \sim 10$ ($V_r \sim 20$) particles. For simplicity, we neglected the difference in materials (i.e., soot or WS) in the potential calculations of Eq. (2), which implicitly assumes that soot material is sufficiently hydrophilic, with high WS wettability. Although we made assumptions in the modeling of these particles, their shapes are similar to those observed by electron microscopy (Adachi et al., 2010; Fig. 5). The shape model of the particle was ultimately defined using a three-dimensional rendering technique.

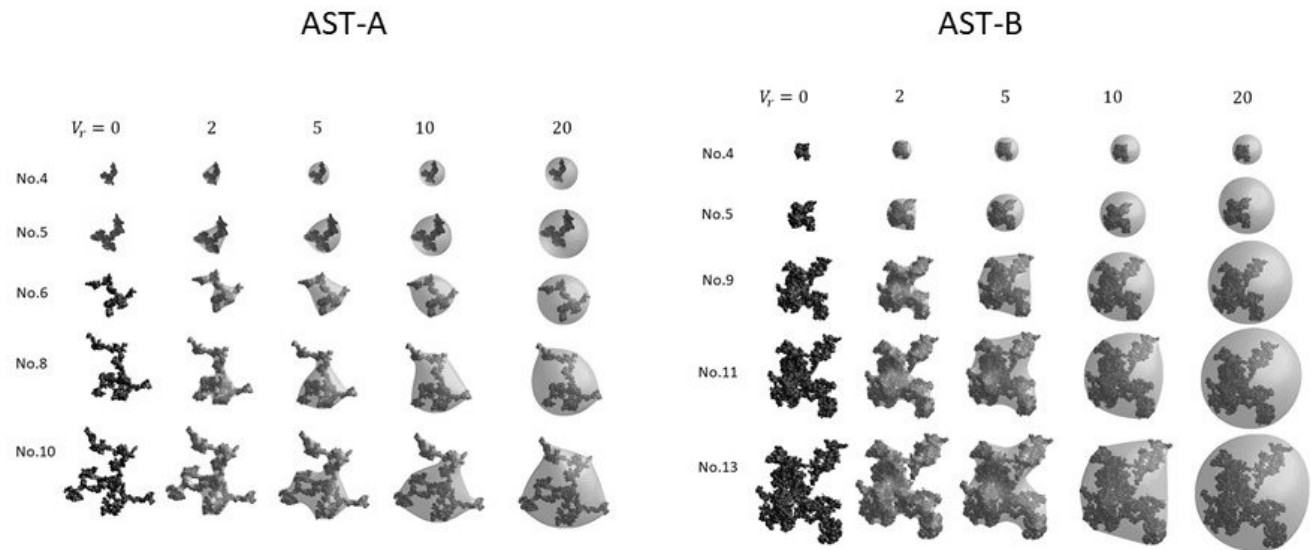

**Figure 4. Mixed soot particle model developed using artificial surface tension (AST) for the attached water-soluble material. Several Type A (Type B) aggregates of different sizes were used on the left (right) side. The marching cubes method was applied for surface rendering.**

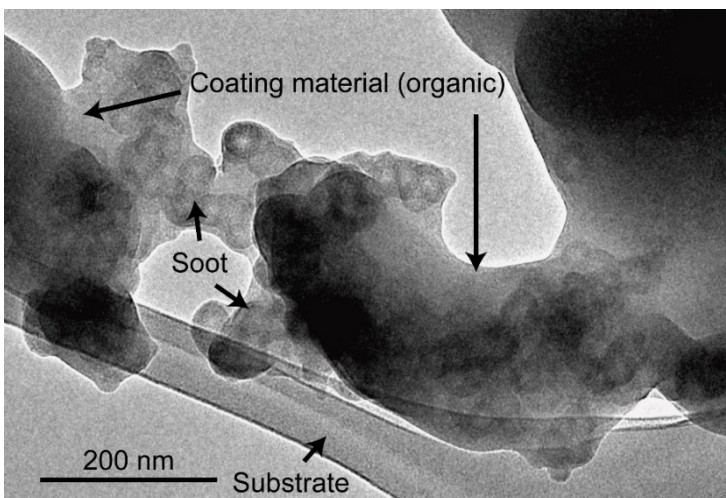

Figure 5. An example transmission electron microscopy (TEM) image of an internally mixed soot particle. The sample was collected from biomass burned during the Biomass Burning Observation Project (BBOP)(Adachi et al., 2018). The soot particle has an aggregate structure of spherical monomers and is embedded within organic material. This particle looks similar to the AST-A particles (e.g., No. 6 particle with $V_r = 5$ in Figure 4).

## 3 Single-scattering properties

We calculated the light-scattering properties using the finite-difference time-domain (FDTD) method (Cole, 2005)(Taflove and Hagness, 2005) (Ishimoto et al., 2012a) and DDA [DDSCAT ver.7.3, http://www.ddscat.org/ (Draine and Flatau, 1994) ]. Particles defined by discrete points as the input for the light-scattering calculations were reconstructed from the shape model described in Section 2, and the grid length was set within the range $2 \leq 2a/l \leq 10$ for correct reproduction of the defined aggregates within the numerical convergence criterion. Although soot particles are generally small aerosols, large computational resources for FDTD and DDA calculations are necessary to estimate the exact scattering properties of the mixed particles due to the fine structure of soot and additional volume contributed by WS material. Using the same discretized particle under an appropriate convergence condition, the scattering properties calculated by FDTD would have approximately the same accuracy as those derived from DDA (Yurkin et al., 2007). However, the numerical cost of the two methods differs depending on the particle shape and the refractive index of the particle material (Yurkin et al., 2007). The memory and CPU time of the DDA calculations mainly depend on the number of discretized points (i.e., dipoles) and depend on particle volume. Meanwhile, the numerical costs of the FDTD calculations are determined by the size of the discretized numerical field that encloses the particle. For fractal-shaped particles of relatively small fractal dimensions, DDA calculations are faster than FDTD ones because of the small relative volume with respect to size. By contrast, the numerical cost of DDA calculations drastically increases as the volume of attached WS elements increases, whereas the FDTD method can output results within similar CPU times given similar total sizes. Therefore, we used both numerical methods for the light-scattering calculations in accordance with the size and shape of the particles. In the numerical environment of our non-parallel computation, light scattering calculations performed using FDTD were faster than those using DDA for AST-B particles (Nos.10–13).

As indicated in Section 2 and shown in Figures 1 and 2, 10 sizes of bare soot aggregates with a volume-equivalent sphere radius $r_{eq} = 0.02 - 0.20 \, \mu m$ for Type-A and 13 sizes with $r_{eq} = 0.02 - 0.42 \, \mu m$ for Type-B were prepared, and internally mixed particles of $V_r \sim 0, 2, 5, 10, 20$ were numerically created for each soot particle. The corresponding size ranges of the mixed soot particles were $r_{eq} = 0.02 - 0.55 \, \mu m$ for Type-A (AST-A) and $r_{eq} = 0.02 - 1.15 \, \mu m$ for Type-B (AST-B). Assuming a synthetic analysis using satellite-/ground-based multi-channel radiometer and lidar measurements, 10 wavelengths

from near-ultraviolet to near-infrared (340, 355, 380, 400, 500, 532, 675, 870, 1020, 1064 nm) were selected. We used a spectral refractive index dataset by (Chang and Charalampopoulos, 1990) for the bare soot material. For the WS components, the dependence of the refractive index on relative humidity was considered, and relative humidity values of 0%, 50%, 90%, and 98% in the software package Optical Properties of Aerosols and Clouds (OPAC) were applied (Hess et al., 1998). The outputs included the results of the light-scattering properties with those averaged over 88 orientations for the FDTD method and 100 orientations for the DDA method.

As examples of the numerical results, Figures 6 and 7 present the size ($r_{eq}$) dependence of single-scattering albedo ($\omega$) and the asymmetry factor ($g$) at wavelength of $\lambda = 532$ nm for AST-A. The complex refractive indices for soot and WS are listed in Table 1, where the refractive index at a relative humidity of 50% were applied for WS. For comparison, the results of $\omega$ and $g$ with the same $V_r$ but derived via Mie calculations for spheres of the effective refractive index calculated using the Maxwell–Garnett mixing rule (MG) (Bohren and Huffman, 1983) and for spheres with a soot-core/WS-shell structure (CS) were also plotted.

For $\omega$ and $g$ at $\lambda = 532$ nm, the results of the AST model were approximately consistent with previous modeled results (Dong et al., 2015)(Liu et al., 2016). The results of $\omega$ for AST-A at $V_r = 0$ markedly differed from those of MG/CS due to the volume-equivalent sphere approximation adopted in the MG/CS treatments. The results of $\omega$ for AST-A with $V_r \geq 2$ showed a similar trend to the MG/CS results such that the MG results were closer to AST-A than was CS. Regarding the asymmetry factor, $g$ depends mainly on the particle size $r_{eq}$ and is less sensitive to the mixing ratio $V_r$. The derived $g$ for AST-A fell between the results of MG and of CS, with the MG results closer to those of AST-A. Because the primary particle of the assumed soot ($a = 0.02$ μm) was smaller than the wavelength and the Type A aggregates were fractal shapes with relatively small $D_f$ (Figs. 1 and 2), the effective medium theory based on the MG mixing rule offered a better approximation than did the CS approximation for the AST-A model.

**Table 1. Complex refractive index ($n + i\,k$) of soot and water-soluble components (WS) used for light scattering calculations (Figs. 6–10). A relative humidity of 50% was assumed for the WS component.**

| wavelength (nm) | soot | | water soluble | |
|---|---|---|---|---|
| | $n$ | $k$ | $n$ | $k$ |
| 355 | 1.392 | 0.6985 | 1.441 | $2.469 \times 10^{-3}$ |
| 532 | 1.723 | 0.5837 | 1.437 | $2.982 \times 10^{-3}$ |
| 1064 | 1.830 | 0.5573 | 1.427 | $8.691 \times 10^{-3}$ |

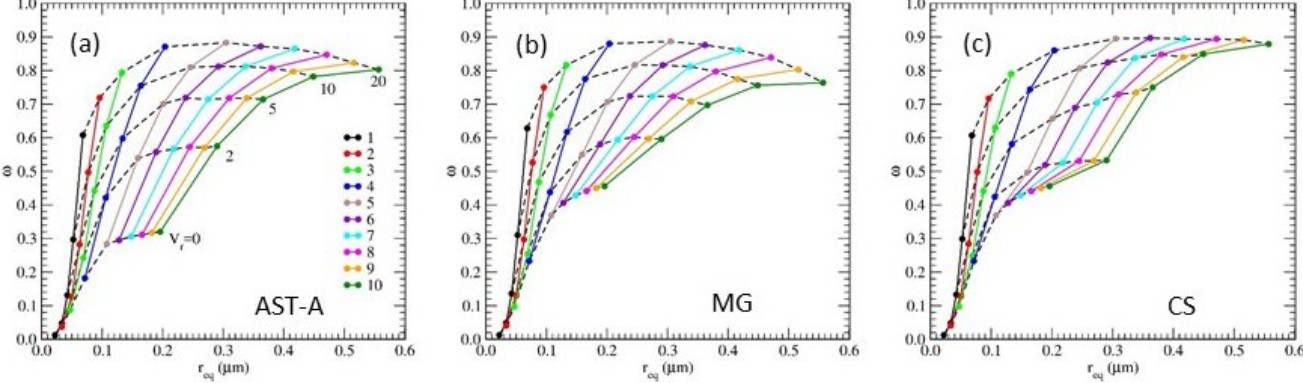

**Figure 6. Single-scattering albedo $\omega$ versus particle size (volume-equivalent sphere radius: $r_{eq}$) at a wavelength of $\lambda = 532$ nm for the (a) AST model of Type A aggregates (AST-A), (b) Maxwell–Garnett approximation (MG), and (c) core-shell approximation (CS). Results for the same bare soot aggregate at different volume ratios of $V_r \sim 0, 2, 5, 10, 20$ are plotted in the same color. Aggregate numbers are the same as those in Figure 1.**

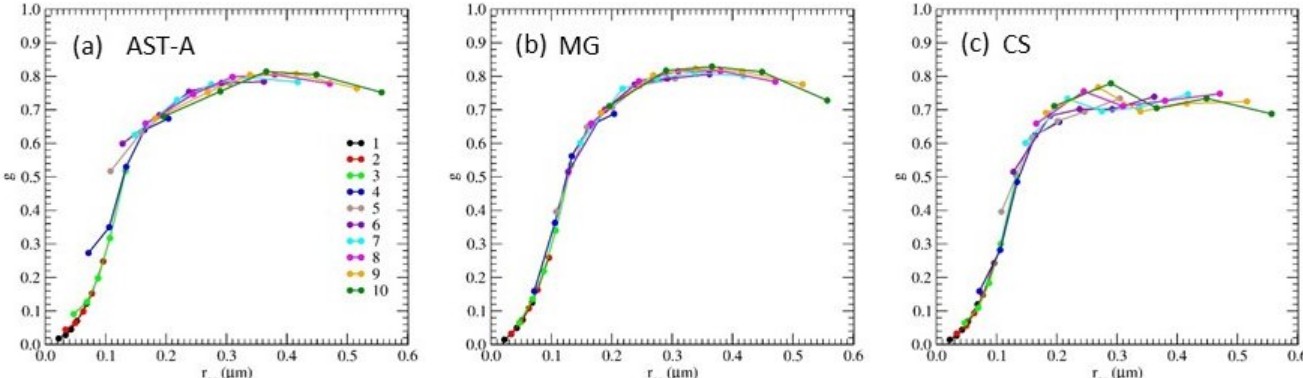

**Figure 7. As Fig. 6, but for the results of the asymmetry factor $g$.**

Compared to $\omega$ and $g$, the backscattering properties of particles are sensitive to particle shape and mixing state. Therefore, lidar measurements could potentially offer information on the validity of the particle model. The calculated lidar ratio $L$ and linear depolarization ratio $\delta_L$ for the AST-A and AST-B particles at wavelengths of 355, 532, and 1064 nm are plotted in Figures 8–10. For a single particle, $L$ is calculated from $\omega$ and the normalized phase function $P_{11}$ in the backscattering direction, and $\delta_L$ is derived from the $P_{11}$ and $P_{22}$ components of the scattering matrix. Here, we omitted the backscattering of $P_{12}$ for $\delta_L$ due to the assumption of random particle orientation.

$$L = \frac{4\pi}{\omega P_{11}}, \delta_L = \frac{P_{11}-P_{22}}{P_{11}+P_{22}}. \qquad (6)$$

For lidar ratios, the MG and CS sphere results for the same volume of bare soot aggregates and the same $r_{eq}$ range of $V_r \leq 20$ as that for AST-A and AST-B are also plotted in Figures 8 and 9. Because backscattering $P_{11}$ is sensitive to the size for spherical particles, calculations were performed at a step size of 0.01 µm. The lidar ratios of CS particles tended to be smaller than those of MG particles, and the difference between MG and CS was significant at wavelengths of 355 nm and 532 nm. Furthermore, the lidar ratios of AST-A and AST-B particles were approximately between the MG and CS results for particles of the same $r_{eq}$ and mixing ratios, particularly for entirely encapsulated $V_r \geq 10$ particles. As denoted in the asymmetry factor results, MG and CS corresponded to two extreme cases of mixture soot materials within the WS shell; our lidar ratio results for fractal-like soot shapes among AST-A and AST-B are reasonable.

Due to strong absorption properties, bare soot particles showed small depolarization ratios $\delta_L$ (Fig. 10). As the value $V_r$ increased, $\delta_L$ of the AST particles increased through the mixing of weak absorbing material (i.e., WS), and $\delta_L$ began to decrease for larger values of $V_r$ due to their spherical shape. This indicates that the smooth surface created by a thin WS coating was rather ineffective for reducing $\delta_L$ when the overall shape was highly non-spherical. As a result, AST particles of the same bare soot size as those shown in Figure 10 (data of same color) had a peak depolarization ratio. To some extent, peak values of $\delta_L$ were larger for larger bare soot particles at shorter wavelengths. This implies that the particle size parameter $x_{eq}$ ($= 2\pi r_{eq}/\lambda$) is an important factor for the evaluation of $\delta_L$ as well as shape non-sphericity.

According to the biomass smoke measurement results, the size of dry particles was 0.10–0.16 µm at count median diameter (0.25–0.30 µm at volume median diameter) (Reid et al., 2005a). Laboratory and *in situ* measurements for aged biomass smoke

yielded estimates of $\omega \geq 0.75$ and $g \sim 0.6$ at $\lambda = 532$ nm (Reid et al., 2005b)(Pokhrel et al., 2016). For lidar measurements, typical values of $L \sim 70$ sr and $\delta_L \sim 7$ % at $\lambda = 532$ nm have been reported (Groß et al., 2015a)(Groß et al., 2015b). Among our optical property results for AST-A, particle No. 4 ($V_r = 20$) had values of $(\omega, g, L, \delta_L) = (0.87, 0.67, 69 \text{ sr}, 0 \text{ %})$, and particle No.5 ($V_r = 2$) had values of $(\omega, g, L, \delta_L) = (0.54, 0.63, 188 \text{ sr}, 6 \text{ %})$. Using the refractive index of a typical soot

particle, relatively high values of $V_r$ would be necessary to simulate $\omega \geq 0.75$. In the AST model, the soot aggregate was entirely encapsulated by WS components, and the overall shape became spherical under large $V_r$ values. Although such spherical particles have consistent lidar ratios of $L \sim 70$ sr, the depolarization ratio becomes $\delta_L \sim 0$ for spherical shapes. By contrast, large depolarization ratios can occur if internally mixed particles are highly non-spherical. Moreover, the reported spectral dependence of depolarization ratios $(\delta_{L,355nm} \sim 20 \text{ %}, \ \delta_{L,532nm} \sim 9 \text{ %}, \delta_{L,1064nm} \sim 2 \text{ %})$ from airborne measurements

for smoke plumes (Burton et al., 2015) (Mishchenko et al., 2016) can be explained, for example, by the AST-A particle No. 6 ($V_r = 5 - 10$; Fig. 10a–c). However, such non-spherical mixed particles tended to have relatively large lidar ratios of $L \geq$ 100 sr at $\lambda = 355, 532$ nm in the AST-A model. In the comparison between AST-A and AST-B particles, variation in $L$ and $\delta_L$ for AST-B were greater than those for AST-A. At $\lambda = 355$ nm, the measurement-derived lidar ratio was $L_{355nm} = 76 \pm$ 12 sr (Groß et al., 2015b). The value was difficult to explain for MG particles, but was relatively consistent with the large $V_r$

observed for AST-A/-B particles and some CS particles. By contrast, the calculated depolarization ratios of AST-A and AST-B particles at both $\lambda = 355$ nm and 532 nm exceeded 30% for larger aggregates with a high amount of coating. These ratios were higher than those measured for biomass burning [7~16% (Groß et al., 2015b)], suggesting that the depolarization ratios of non-spherical soot particles were easily enhanced when coated with weakly absorbing WS material. A similar effect was previously reported (Kahnert, 2017). According to (Kanngießer and Kahnert, 2018), the speed of transition from film coating

to spherical growth (Pei et al., 2018) is a morphological parameter that strongly affects the depolarization ratio. A rapid transition to spherical growth may cause a smaller depolarization ratio of internally mixed soot particles.

Overall, the results shown in Figures 8–10 suggest that the effects of internal mixing on lidar backscattering are strongly related to changes in absorption/shape properties due to mixing and bare soot particle size. However, it was difficult to simulate average smoke optical properties $(\omega, g, L, \delta_L)$ and their spectral dependence solely using AST-A or AST-B particles.

Among the field measurements, the observed optical properties were the average of the particle size distribution. Our results indicate that the presence of large mixed soot particles may enhance the bulk $\delta_L$ of smoke.

Typical values of $V_r$ for mixed biomass smoke particles likely depend on the relative humidity, concentrations of WS components, and size/shape of soot aggregates. The mixing of different aerosol types, such as dust in biomass burning (Groß et al., 2011)(Groß et al., 2013), may be important for the interpretation of measured optical properties. Improved retrieval

calculations based on realistic aerosol simulations that consider particle size/shape distribution and other types of aerosol contamination are expected using our AST model for internally mixed soot particles.

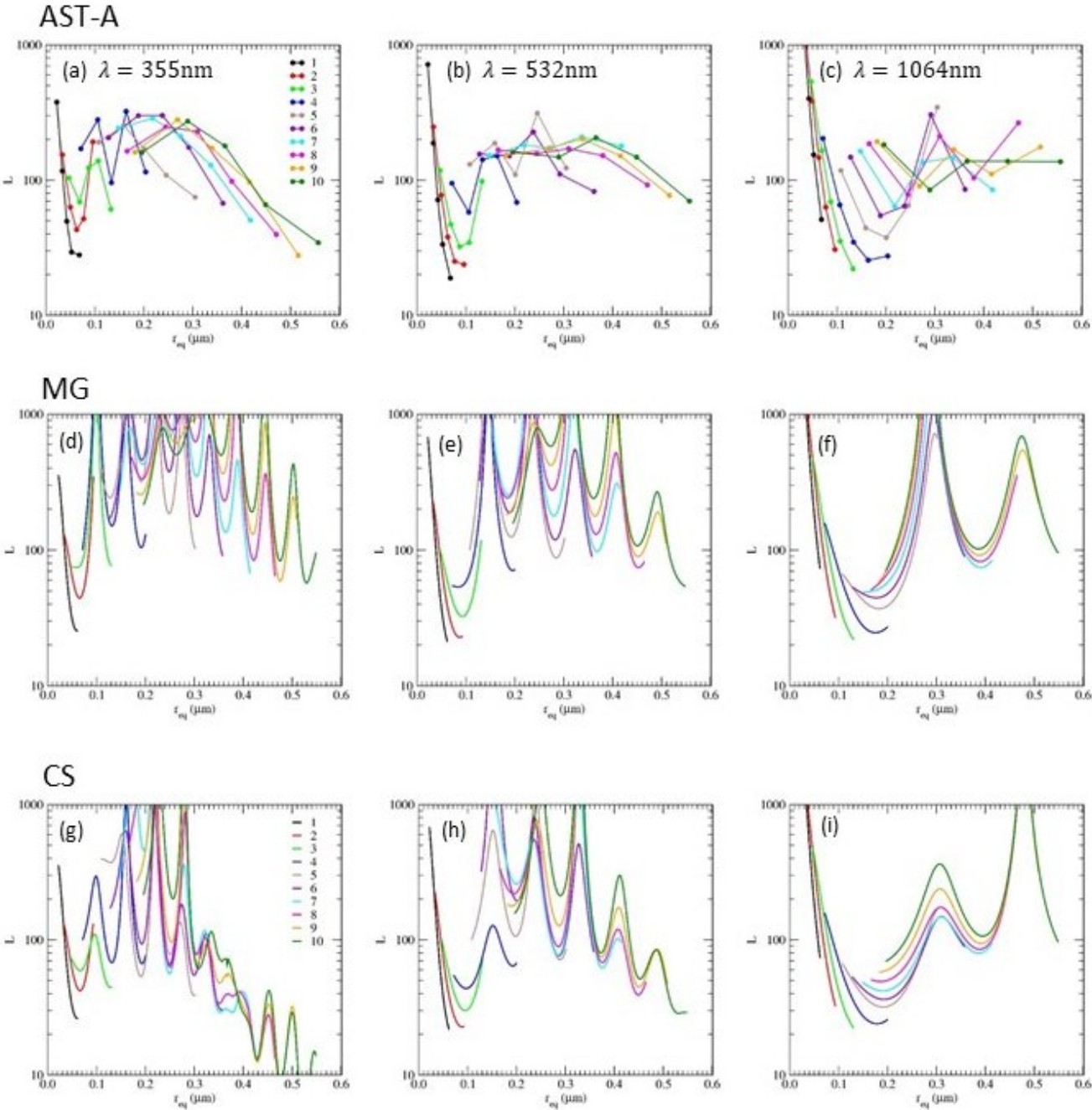

**Figure 8. Results of the lidar ratio _L_ for particles derived from the AST-A model at wavelengths of (a) 355, (b) 532, and (c) 1064 nm. (d–f) As a–c, but for spheres with an average refractive index calculated using the Maxwell–Garnett mixing rule (MG). (g–i) As a–c but for core-shell spheres (CS). Colors and volume ratios $V_r$ for each point are the same as those in Figure 6a for the AST-A model ($V_r{\sim}0, 2, 5, 10, 20$ from left to right for circles of the same color). For (d–i), calculations were performed for $V_r \leq 20$ with a step size of 0.01 µm.**

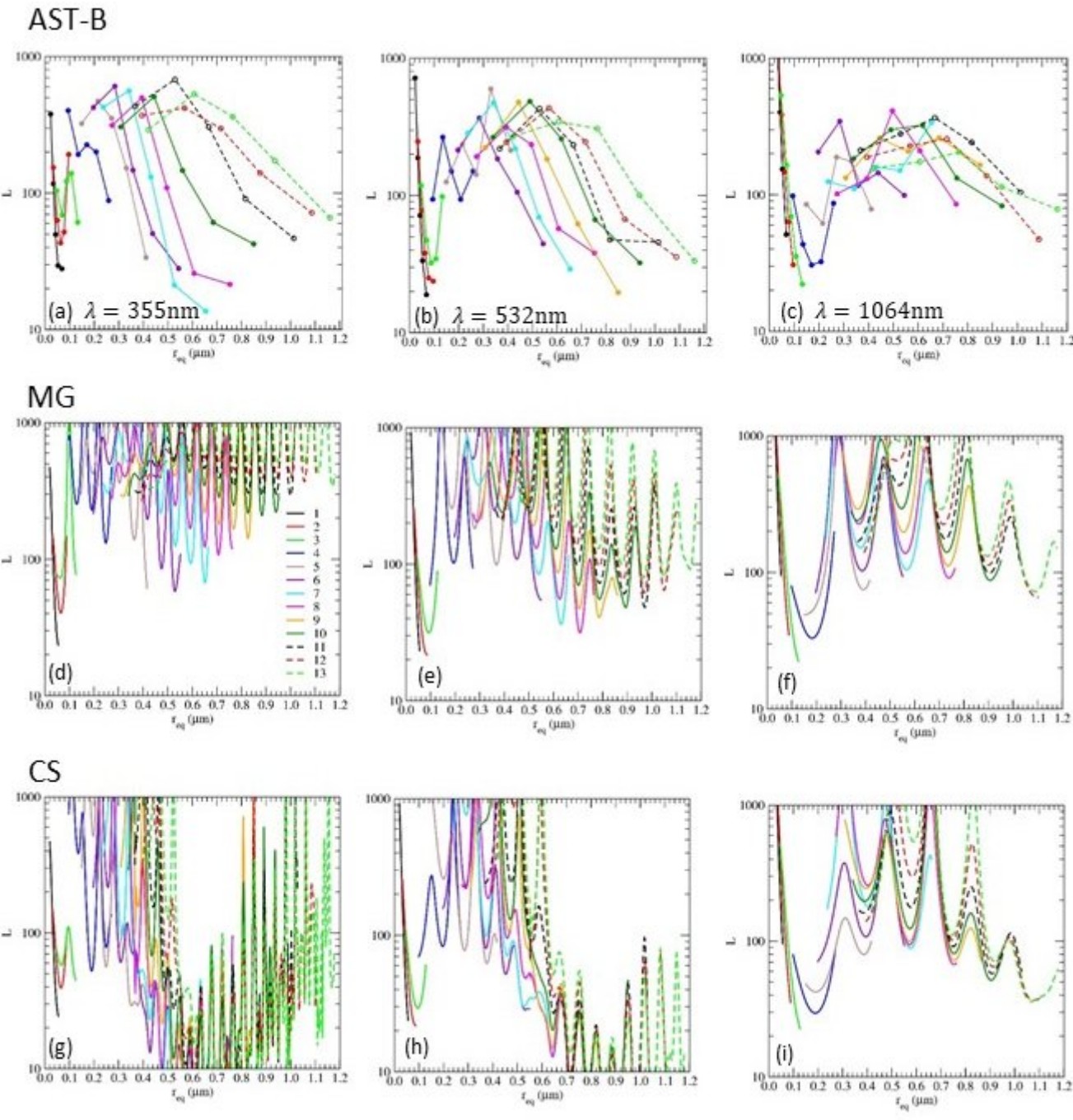

**Figure 9. As Fig. 8, but for internally mixed Type B particles (AST-B) (a–c), MG (d–f), and CS (g–i). The horizontal scale differs from that shown in Figure 8.**

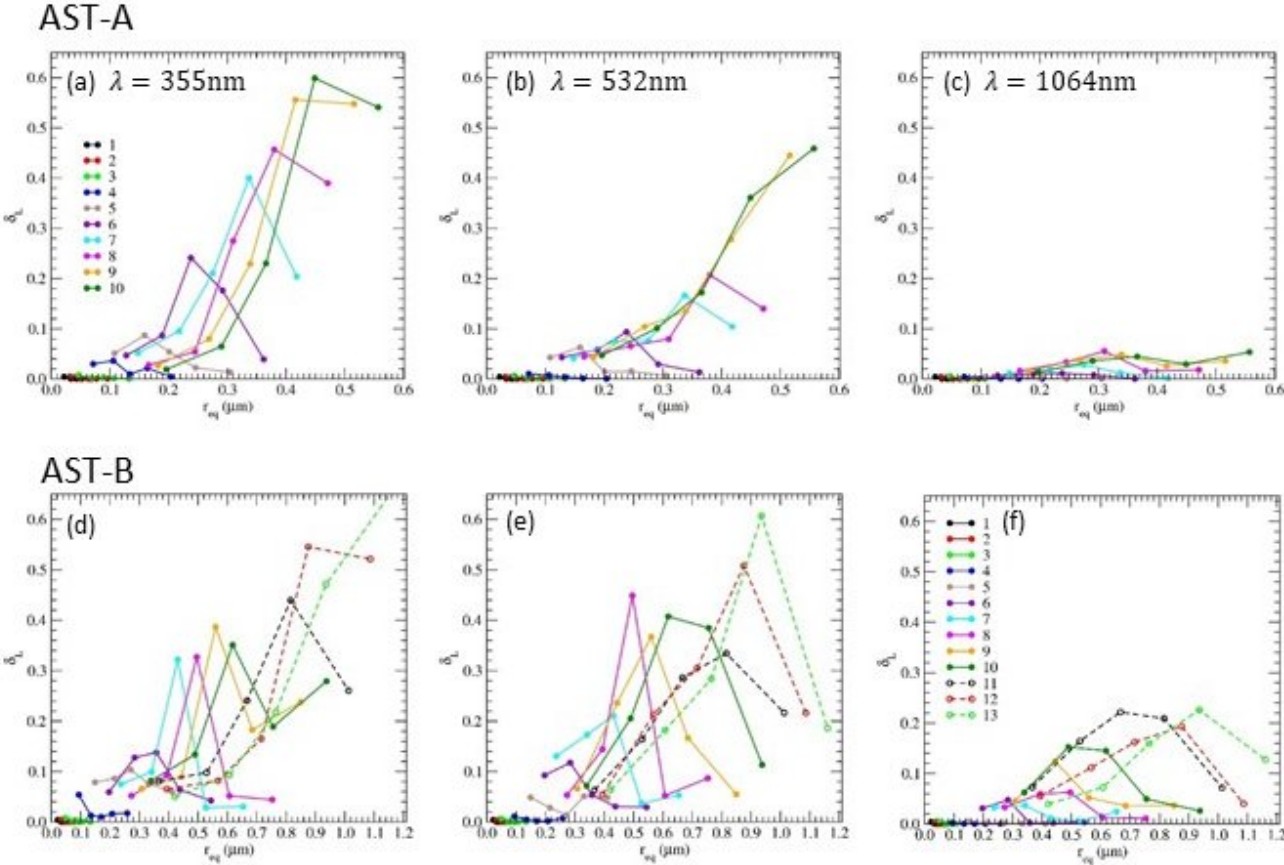

**Figure 10. Results of the linear depolarization ratio $\delta_L$ for particles derived from the AST-A model at wavelengths of (a) 355, (b) 532, and (c) 1064 nm. (d–f) As (a–c), but for AST-B particle results.**

## 5 Summary

For satellite-/ground-based remote-sensing analysis applications, we developed a shape model of internally mixed soot particles and calculated the optical properties of particles at visible and near-infrared wavelengths. Fractal-like structures extracted using spatial Voronoi tessellation were considered to mimic necking and overlapping between neighboring primary particles. We created two types of polyhedral aggregates with different particle shape–size dependences (Type-A and Type-B) to account for the effects of compaction on soot aggregate shape during the atmospheric aging process. Then the artificial surface potential for the particle in a Cartesian grid space was defined, and the surface tension of the WS components on the soot aggregate was simulated, assuming that the soot was hydrophilic with high wettability. Based on a simple assumption of the behavior of WS elements, the shapes of internally mixed soot particles dependent on the amount of attached dissolved material were determined from iterative calculations. Overall, an aggregate coated with a thin film was simulated given a relatively small volume of added WS components, whereas the soot aggregate was covered with a spherical shell given a large amount of WS components. The optical properties of the developed internally mixed particles were calculated using the FDTD and DDA methods while considering the spectral dependence on the refractive indices.

For single-scattering albedo and asymmetry, AST model results were similar to those for MG and CS, except when the WS ratio was low ($V_r \sim 0$). Due to shape irregularities, the lidar ratios of AST particles in random orientation were less sensitive to particle size than those of MS and CS particles. The $L$ values of AST particles were approximately between those of MG and CS particles. For AST particles, $\delta_L$ increased as the amount of weakly absorbing material (i.e., WS) increased, and $\delta_L$
decreased as particle shape became spherical. As a result, irregularly shaped soot particles had peak depolarization ratios during the internal mixing process. Maximum $\delta_L$ values tended to be larger when the size parameters of non-spherical mixed particles were large. Following comparisons with reported optical features, we determined that average optical properties ($\omega, g, L, \delta_L$) and their spectral dependences for measured biomass burning aerosols cannot be simulated by a single particle in our internally mixed soot model. In particular, AST particles tended to have larger lidar ratios and depolarization ratios than
those obtained by field measurements. The inconsistencies in lidar backscattering properties between the model results and measurements may remain unresolved even after size-averaging the optical properties. One possible explanation for this phenomenon is that soot aggregates change their shapes to become more compact (i.e., become spherical) during the WS adhesion process in the atmosphere. The contribution of the core shell type of internally mixed soot particles and the use of the non-spherical model may be necessary to simulate observed results for burned biomass aerosols.

In addition to its use in direct analyses for the field measurement results, the dataset of the AST particle optical properties can be used for the parameterization of conventional spherical particle models, such as the MG model, CS model, and their combined model. Furthermore, the dataset will be useful for determining the shape property conditions of smoke particles observed from multi-sensor measurements, including lidar backscattering.


**Acknowledgements**

This work was supported by the Global Change Observation Mission-Climate (GCOM-C) research project of Japan Aerospace Exploration Agency (JAXA), and by the Global Environmental Research Fund of the Ministry of the Environment (MOE) in Japan.

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
