# Peer review of "A shape model of internally mixed soot particles derived from artificial surface tension"

_Atmospheric Measurement Techniques, 2018_

## Referee Comment (RC1) · Anonymous Referee #1 · 19 Sep 2018

General comments: This paper describes shape models for soot and internally mixed soot and their optical characteristics. This kind of work is important and essential to understand microphysics and to analyze observations correctly. In that sense, the paper is suitable for AMT. The paper is well written, generally.

Specific comments: The descriptions on the iteration procedure for calculating mixed soot particles are difficult to understand. How many iterations are necessary to obtain Wr=20? What is the relationship between relative humidity and Wr? What is the relative humidity in Figs. 7-9?

Lidar ratio values should be discussed in more details. Lidar ratio values at 355 nm and 532 nm reported in observational studies should be summarized. In my understanding, the observed lidar ratio at 355 nm was smaller than 532 nm in forest fire cases. It looks

the wavelength dependence in Fig. 8 is opposite for small Wr. However, it looks good for large Wr, for example, A-7 Wr=20. The depolarization ratios are also close to the observation, in this case. Considering the lidar ratio values, it may be more appropriate to consider large Wr, even if the depolarization ratios are not well reproduced.

Lidar ratios calculated with MG and CS should be also presented.

————————————————

---

## Short Comment (SC1) · 26 Sep 2018

Ishimoto et al. presented a very interesting shape model of coated soot particles. This comment is mainly referring to the high depolarisation ratios obtained with the presented shape model.

For larger aggregates with a high amount of coating the calculated depolarisation ratios for both 355 and 532 nm exceed 0.3, which appears to be too high for soot particles. Some of the modelled particles even give depolarisation ratios, which are higher than 0.5, which is higher than what would be expected of any type of atmospheric aerosol. When considering aerosol size distributions for remote sensing applications these high values pose the risk of overestimating the depolarisation ratios with your shape model.

[Figure]

A similar effect has been reported by Kahnert, 2017. A possible explanation was that the coated soot particles were too non-spherical. Your results point into a similar direction, as some of the soot aggregates with Vr=20 have a smaller depolarisation ratios than the same aggregates with Vr=10. In Kanngießer and Kahnert, 2018 we showed that the speed of transition from film coating to spherical growth is a morphological parameter which has a strong impact on the depolarisation ratio. Having a faster transition to spherical growth resulted in more spherical paticles which had a smaller depolarisation ratio. Does your model allow for a more spherical shape for coatings of larger aggregates with e.g. Vr=10?

The lab experiments conducted by Pei et al., 2018 suggest that with the application of coating material the soot aggregates become compacted before there will be a growth in particle size. Could the different particle types (AST-A and AST-B) represent such compaction?

Kahnert, M., "Optical properties of black carbon aerosols encapsulated in a shell of sulfate: comparison of the closed cell model with a coated aggregate model," Opt. Express, 25, 24579-24593, https://doi.org/10.1364/OE.25.024579, 2017

Kanngießer, F, Kahnert, M., Calculation of optical properties of light-absorbing carbon with weakly absorbing coating: A model with tunable transition from film-coating to spherical-shell coating, J. Quant. Spectrosc. Radiat. Transfer, Volume 216, Pages 17-36, https://doi.org/10.1016/j.jqsrt.2018.05.014, 2018

Pei, X., Hallquist, M., Eriksson, A. C., Pagels, J., Donahue, N. M., Mentel, T., Svenningsson, B., Brune, W., and Pathak, R. K.: Morphological transformation of soot: investigation of microphysical processes during the condensation of sulfuric acid and limonene ozonolysis product vapors, Atmos. Chem. Phys., 18, 9845-9860, https://doi.org/10.5194/acp-18-9845-2018, 2018.

---

## Referee Comment (RC2) · Anonymous Referee #2 · 30 Sep 2018

This manuscript presents optical modeling of internally mixed soot particles, a subject that is of interest for atmospheric remote sensing. The modeling approach and computational techniques are sound and reasonable. The results are representative and semi-extensive, and thus should be useful in remote sensing analysis and relevant optical interpretation. I suggest the following revisions for the authors' consideration:

1) Is Eq. (1) necessary? Remove it, if it is unnecessary.

2) The discussion on the efficiency of computational methods only focuses on the shape aspect. Actually, the refractive index has large impact on the efficiency comparison between FDTD and DDA.

3) It might be better to have a table of the refractive indices at the 10 wavelengths.

4) A reference is required for the Maxwell-Garnett mixing rule.

5) The results are presented for single particles. It is unclear to obtain the size-averaged results from the simulated results. More discussion is required on the comparison between simulations and observations/measurements.

6) In summary, it might be better to summarize the new knowledge gained from the present modeling study.

---

## Author Comment (AC1) · 12 Nov 2018

The answer to the reviewer's comments was attached in the form of a supplement.

Please also note the supplement to this comment:
https://www.atmos-meas-tech-discuss.net/amt-2018-249/amt-2018-249-AC1-supplement.zip

---

## Author Response (AR1)

Response to reviewer (comment-to-author)

We thank the reviewer for taking the time to provide us with helpful comments that we believe have substantially improved our paper. We address each concern of the reviewer on a point-by-point basis as follows:

1) Eqs. (3): the right hand side does not include "j" which appears in the left hand side. Maybe "ri" means "rij". Please improved the expression.

Reply: The expression was corrected (Page 4 Line 11)

2) Eqs. (4)-(6): The equations are hypothetical and the physical meanings are not well described. For example, what is the basis of assumption of "d1" can be defined by three times larger than the grid size "l", even when there is no specification of the value of "l" in the manuscript. This editor feels such hypothetical relations can be introduced in a paper in the situation of no past similar studies are there, but at least the authors should put more explanation, especially for physical meanings of the assumption.

Reply: Some sentences were added for the background and explanation regarding physical meaning of our approach (Page 4 Line 1-4) and some sentences were modified (Page 4 Line 8-9, 24-26).

3) page 8, line 10 (30?): It will be useful for reader to see the value of the complex refractive index at 532nm with those values at other two wavelengths, because the results are sometimes given for 532nm in Figs. 5-9.

Reply: A table for refractive index was added (Table 1).

4) Figs. 6, 8: It will be useful to show at which size the DDA is connected with FDTD to study the consistency of the computed values by the two methods.

Reply: In the numerical environment of our non-parallel computation, light scattering calculations performed using FDTD were faster than those using DDA for AST-B particles (Nos.10–13). The same sentence was added in the text (Page 6 Line 27-28).

**5) Also equivalent homogeneously mixed sphere case can be added for how the internal mixture of BC and WS is different from the usual Mie sphere case.**

Reply: For comparison, results of MG and CS for lidar ratios were added (Fig. 8-9).

**6) page 10: A table to summarize the values of optical cross sections will be useful for readers to see dependences of discussed parameters on the wavelengths.**

Reply: Although we tried to make the table, we gave up because it would be a very large table. For example, 10 (bare soot size) x 5 (mixing state) x 3 (wavelength) x 2 (cross sections Cext, Cabs) = 300 data is necessary for the table of type-A particles. For type-B particles, another table of 390 data is needed. We would like to upload the data files as a supplement together with our final manuscript.

Response to reviewer (RC1)

We thank the reviewer for taking the time to provide us with helpful comments that we believe have substantially improved our paper. We address each concern of the reviewer on a point-by-point basis as follows:

**1) The descriptions on the iteration procedure for calculating mixed soot particles are difficult to understand. How many iterations are necessary to obtain Wr=20? What is the relationship between relative humidity and Wr? What is the relative humidity in Figs. 7-9?**

Reply: To produce $V_r \sim 20$ particles, we started from the state of bare soot ($V_r = 0$) and the result of particle shape was outputted when $V_r$ exceeded 20. As described in Eq. (3), ~2 % of the surface points defined in the grid space are adhered as WS material for one iteration (including two steps). Approximately, 1000 (2000) iterations were applied for making the $V_r \sim 10$ ($V_r \sim 20$) particle. A sentence was added in the text (Page 5 Line 13-14). The relative humidity is a parameter to determine the refractive index of water soluble (WS). We used the results of refractive index at relative humidity 50% for the plots of section 3. For the dataset of light scattering properties, we calculated the optical properties of particles for 4 cases of relative humidity (0%, 50%, 90%, 98%). The values of relative humidity and refractive indices were mentioned in the text (Page 7 Line 8-9) and in the caption of Table 1.

**2) Lidar ratio values should be discussed in more details. Lidar ratio values at 355 nm and 532 nm reported in observational studies should be summarized. In my understanding, the observed lidar ratio at 355 nm was smaller than 532 nm in forest fire cases. It looks the wavelength dependence in Fig. 8 is opposite for small Wr. However, it looks good for large Wr, for example, A-7 Wr=20. The depolarization ratios are also close to the observation, in this case. Considering the lidar ratio values, it may be more appropriate to consider large Wr, even if the depolarization ratios are not well reproduced. Lidar ratios calculated with MG and CS should be also presented.**

Reply: Results of lidar ratios for MG and CS were added (Fig.8-9). We also

added a short discussion about the results of lidar ratio (Page 8 Line 18-25, Page 9 Line 13-21) including the results of depolarization ratios. As the reviewer pointed out, lidar ratio at 355 nm can be smaller than that at 532 nm depending on the size and mixing condition of the particle.

Response to reviewer (RC2)

We thank the reviewer for taking the time to provide us with helpful comments that we believe have substantially improved our paper. We address each concern of the reviewer on a point-by-point basis as follows:

**1) Is Eq. (1) necessary? Remove it, if it is unnecessary.**

Reply: The equation was removed from the manuscript.

**2) The discussion on the efficiency of computational methods only focuses on the shape aspect. Actually, the refractive index has large impact on the efficiency comparison between FDTD and DDA.**

Reply: A sentence was modified and reference Yurkin et al. (2007) was added (Page 6 Line 19-20).

**3) It might be better to have a table of the refractive indices at the 10 wavelengths.**

Reply: A table for the refractive index was added (Table 1). Because we only showed the plots for the results of 3 wavelengths (wavelengths for lidar measurements) with one relative humidity (50 %) for water soluble, the refractive indices were listed only for the corresponding 3 wavelengths to avoid the reader's confusion.

**4) A reference is required for the Maxwell-Garnett mixing rule.**

Reply: Reference was added (Page 7 Line 11).

**5) The results are presented for single particles. It is unclear to obtain the size averaged results from the simulated results. More discussion is required on the comparison between simulations and observations/measurements.**

Reply: In the revised manuscript, we expanded the part of discussion in

section 3 and in summary. We are preparing another paper regarding the retrieval of soot particles from satellite measurements by using size averaged optical properties. We would like to discuss this issue in our future work if we obtain the reviewer's approval.

**6) In summary, it might be better to summarize the new knowledge gained from the present modeling study.**

Reply: Our new findings about single scattering properties were summarized (Page 13 Line 1-14).

Response to reviewer (SC1)

We thank the reviewer for taking the time to provide us with helpful comments that we believe have substantially improved our paper.

• For larger aggregates with a high amount of coating the calculated depolarisation ratios for both 355 and 532 nm exceed 0.3, which appears to be too high for soot particles. Some of the modelled particles even give depolarisation ratios, which are higher than 0.5, which is higher than what would be expected of any type of atmospheric aerosol.
• When considering aerosol size distributions for remote sensing applications these high values pose the risk of overestimating the depolarisation ratios with your shape model. A similar effect has been reported by Kahnert, 2017.
• A possible explanation was that the coated soot particles were too non-spherical. Your results point into a similar direction, as some of the soot aggregates with Vr=20 have a smaller depolarisation ratios than the same aggregates with Vr=10. In Kanngießer and Kahnert, 2018 we showed that the speed of transition from film coating to spherical growth is a morphological parameter which has a strong impact on the depolarisation ratio. Having a faster transition to spherical growth resulted in more spherical paticles which had a smaller depolarisation ratio.

Reply: Many thanks for very important comments and information. We essentially agree with the reviewer's opinion. In the revised manuscript, sentences of the same meaning and references were added (Page 9 Line 15-21, Page 13 Line 9-14).

Does your model allow for a more spherical shape for coatings of larger aggregates with e.g. Vr=10? The lab experiments conducted by Pei et al., 2018 suggest that with the application of coating material the soot aggregates become compacted before there will be a growth in particle size. Could the different particle types (AST-A and AST-B) represent such compaction?

Reply: In the revised manuscript, results of lidar ratios for Maxwell-Garnett

(MG) spheres and Core-Shell (CS) spheres were included (Figs 8-9). Approximately, lidar ratios for AST-A and -B particles were in between the results of MG and CS. The modeling and light scattering calculations for more compact soot aggregates than Type-B are possible. We think that the results will be similar to those of CS model. However, we did not carry out these calculations because the results of backscattering properties will show the same fluctuating feature as those of CS particles (Fig. 8-9), and because we have to implement many DDA/FDTD calculations to show general tendency.

[revised manuscript text omitted]